# DC Voltage Sensorless Predictive Control of a High-Efficiency PFC Single-Phase Rectifier Based on the Versatile Buck-Boost Converter

**DOI:** 10.3390/s21155107

**Published:** 2021-07-28

**Authors:** Catalina González-Castaño, Carlos Restrepo, Fredy Sanz, Andrii Chub, Roberto Giral

**Affiliations:** 1Facultad de Ingeniería, Ingeniería Mecatrónica de la Universidad Manuela Beltrán, Bogotá 110231, Colombia; catalina.castano@docentes.umb.edu.co (C.G.-C.); fredy.sanz@umb.edu.co (F.S.); 2Department of Electromechanics and Energy Conversion, Universidad de Talca, Curicó 3340000, Chile; 3Department of Electrical Power Engineering and Mechatronics, Tallinn University of Technology, 19086 Tallinn, Estonia; andrii.chub@taltech.ee; 4Departament d’Enginyeria Electrònica, Elèctrica i Automàtica, Escola Tècnica Superior d’Enginyeria, Universitat Rovira i Virgili, 43007 Tarragona, Spain; roberto.giral@urv.cat

**Keywords:** AC-DC conversion, sensorless, predictive control, buck-boost converter, high efficiency conversion, SEPIC, versatile buck-boost

## Abstract

Many electronic power distribution systems have strong needs for highly efficient AC-DC conversion that can be satisfied by using a buck-boost converter at the core of the power factor correction (PFC) stage. These converters can regulate the input voltage in a wide range with reduced efforts compared to other solutions. As a result, buck-boost converters could potentially improve the efficiency in applications requiring DC voltages lower than the peak grid voltage. This paper compares SEPIC, noninverting, and versatile buck-boost converters as PFC single-phase rectifiers. The converters are designed for an output voltage of 200 V and an rms input voltage of 220 V at 3.2 kW. The PFC uses an inner discrete-time predictive current control loop with an output voltage regulator based on a sensorless strategy. A PLECS thermal simulation is performed to obtain the power conversion efficiency results for the buck-boost converters considered. Thermal simulations show that the versatile buck-boost (VBB) converter, currently unexplored for this application, can provide higher power conversion efficiency than SEPIC and non-inverting buck-boost converters. Finally, a hardware-in-the-loop (HIL) real-time simulation for the VBB converter is performed using a PLECS RT Box 1 device. At the same time, the proposed controller is built and then flashed to a low-cost digital signal controller (DSC), which corresponds to the Texas Instruments LAUNCHXL-F28069M evaluation board. The HIL real-time results verify the correctness of the theoretical analysis and the effectiveness of the proposed architecture to operate with high power conversion efficiency and to regulate the DC output voltage without sensing it while the sinusoidal input current is perfectly in-phase with the grid voltage.

## 1. Introduction

Sensorless control methods are widely used in different fields, among which the regulation of electrical machines stands out [1,2,3]. An AC motor control requires determining the speed and flux position of the motor; however, the mechanical sensor has the disadvantages of high cost, large volume, and poor anti-interference ability caused by temperature and electromagnetic noise. Sensorless control provides a low-cost option, and motors’ sensorless speed control has positioned itself as a relevant alternative from a research and industry perspective [4]. The control of different kinds of motors can be done either with or without sensors. Sensorless control techniques are increasingly used to reduce the overall cost and size of actuating devices in many applications despite requiring more complex control algorithms. The literature presents numerous examples of the sensorless control of electrical machines applied along with estimation of different operating variables (current, voltage, speed, position). In [5], a field-oriented control of a permanent magnet machine drive with an estimation of the currents and voltages of the LC-filter connected in series is presented. In [6], a speed estimation of a linear ultrasonic motor is used to avoid speed/position sensors. In [7], a motion control of a linear resonant actuator with two degrees of freedom is implemented without a position sensor. Another sensorless strategy for speed control of a permanent magnet synchronous motor is explained in [8]. Speed and rotor resistance estimations for an induction motor drive can be found in [9]. Similarly, speed sensorless vector control of an induction motor drive system is implemented in [10]. Finally, a complete review of position and speed sensorless methods for controlling brushless direct current motor drives is included in [11].

Sensorless control methods have also been used in power electronics applications, especially in controlling power factor correction (PFC) single-phase rectifiers [12,13,14,15]. The conventional PFC converter employs a switching power converter between the diode bridge rectifier providing rectified sinusoidal voltage and a DC regulated voltage at its output [13]. In general, the control of a single-phase PFC rectifier requires at least three sensors [15]:
An AC voltage sensor: is employed to detect the phase angle of the source voltage, which is then used to generate the unitary sine current reference for the power factor control;A DC voltage sensor: is used to regulate the converter output voltage and provide load overvoltage protection;A DC current sensor: is utilized to achieve closed-loop power factor control, DC voltage regulation, and overcurrent protection to the load.

There are numerous issues associated with the use of many sensors in the control of PFC single-phase rectifiers, among which the following stand out:Increase of the complexity and cost of the control circuit that is a problem for applications with very stringent cost and space requirements.Reduction of the power converter reliability.Increase in noise coupling and grounding issues of mixed-signal printed circuit board (PCB) from the connection between the power converter and the controller.

Based on the above mentioned considerations, a reduction in the required number of sensors is highly desired in this application. In addition, sensing all the variables in either the input or the output side of the converter stage eliminates the need for galvanically isolated sensors in the control system and reduces the noise coupling between the power converter and the controller. From the three standard sensors of a conventional PFC rectifier, eliminating the DC voltage sensor is the most useful from the viewpoints of cost and reliability [13]. High reliability is desired in commercial single-phase PFC rectifiers, along with High power density and High efficiency (H3) [15]. On the one hand, a high power density is achieved with a high switching frequency that depends on the advance of power electronic components and materials (such as SiC or GaN), packaging techniques, thermal management techniques, and others [16]. On the other hand, the power conversion efficiency depends on the specific converter topology and the operating conditions. According to the literature studies, the primary breakdown of total losses in a power converter is the switching and conduction losses of the transistors and the inductor losses. Other kinds of power losses such as capacitance losses, gate drive losses, printed circuit board layout losses, EMI filter losses, and auxiliary power and controller energy consumption are negligible concerning those mentioned above [17,18,19,20,21,22].

In this paper, a new single-phase PFC rectifier based on the versatile buck-boost converter is proposed. This converter has comparative advantages such as high efficiency, wide bandwidth, smooth transition between buck and boost modes, and the possibility of controlling either input or output voltages or currents that have been addressed in previous works [23,24,25,26]. These relevant advantages tested in DC-DC applications, and especially the high-efficiency property, can be extended to AC-DC applications, which is the primary goal of this article. Furthermore, this paper proposes a fair comparison with two other well-known classical buck-boost converters: the SEPIC [27,28,29,30] and the noninverting buck-boost converter, refs. [31,32] to demonstrate that the proposed versatile converter has a superior efficiency performance in the PFC applications.

Therefore, the main contributions of this paper can be summarized as follows:
A novel PFC single-phase rectifier based on the VBB converter is presented. Its many advantages tested in DC-DC applications are now meant to be proved in an AC-DC PFC application.A high-reliability is obtained by reducing the number of sensors, as critical components in reliability of the converter [33]. This reliability improvement does not compromise the control performance [34]. The ideal voltage conversion ratio of the proposed rectifier converter is used to estimate the DC output voltage. In this way, the controller senses only input variables (DC current and AC voltage), which reduce the noise coupling between the converter input and output terminals.A high-efficiency converter topology operating as PFC single-phase rectifier is achieved. Its power conversion efficiency is compared with other classical PFC rectifiers as the SEPIC and the noninverting buck-boost converter.A converter topology with a wide bandwidth and, consequently, with fast control loops, is proposed. The design of two nested control loops corresponds to a predictive-based current controller (inner loop for the DC input current) and a proportional-integral (PI) voltage controller (outer loop for the DC output voltage). These controllers operate along with output voltage estimation and grid-synchronising phase-locked loop (PLL) running in a low-cost microcontroller. Each of the proposed controllers ensure fast-tracking of the control set-points, and low steady-state error under demanding tests that include changes in the output voltage reference and the resistive load.The theoretical predictions are validated with temporal and thermal simulations as well as with hardware-in-the-loop (HIL) real-time simulation, which has proven to be a handy tool in many applications [35,36,37,38,39,40,41,42].

The structure of this paper is as follows: in Section 2, the dynamic models of three buck-boost converters and the current control approach are presented. The methods used to calculate the losses are described in Section 3. The basic principles of the voltage sensorless control are explained in Section 4. Temporal and thermal simulation and HIL real-time simulation results are included in Section 5. Finally, the conclusions of this work are presented in Section 6.

## 2. Modeling of DC-DC Buck-Boost Converters

Buck-boost converters can be used in PFC single-phase rectifier applications if the output voltage to regulate is lower than the peak AC input voltage. In the following subsections, the dynamic models for the SEPIC, noninverting, and versatile buck-boost DC-DC converters are described to obtain the inductor current slopes. These slope equations are required for the current control design. In addition, information about the selected components and the main parameters of each converter is defined.

### 2.1. SEPIC Converter

Figure 1 shows the circuit topology of the SEPIC converter. The topology consists of two inductors, a capacitor, and two MOSFET switches. The averaged model of the SEPIC converter is as follows [43]:
(1)$$\frac{di_{g}(t)}{dt}=\frac{-u_{2}(v_{c}+v_{o})+V_{g}}{L_{1}}$$
(2)$$\frac{di_{L}(t)}{dt}=\frac{u_{1}v_{c}-u_{2}v_{o}}{L_{2}}$$(3)$$\frac{dv_{c}(t)}{dt}=\frac{u_{2}i_{g}-u_{1}i_{L}}{C}$$(4)$$\frac{dv_{co}\left(t\right)}{dt}=\frac{u_{2}\left(i_{g}+i_{L}\right)}{C_{o}}-\frac{1}{R_{o}\;C_{o}}v_{o}$$

The complementary switches signals are $u_{1}$ and $u_{2}$, and it duty cycle is calculated as follows:(5)$$D=\frac{v_{o}}{v_{g}+v_{o}}$$The inductors $L_{1}$ and $L_{2}$ are selected to obtain a ripple current of 2 A for $v_{g}=200$ V and $v_{o}=300$ V in boost mode (D > 0.5) according to the following expression and $L_{s}=L_{1}=L_{2}$:
(6)$$\Delta i_{L}=\frac{DV_{g}}{L_{s}f_{s}}.$$Therefore, the selected value for both inductors $L_{1}$ and $L_{2}$ is 600 μH for a switching frequency of 100 kHz. The capacitor value is selected for a voltage ripple $\Delta v_{c}=6$ V at the voltage operation point ($v_{g}$ = 200 V and $v_{o}$ = 300 V), with an output current load of 8 A, and following the design expression (Equation 7). Therefore, the value for the capacitor selected corresponds to *C* = 8 μF.
(7)$$C=\frac{i_{o}D}{\Delta v_{c}f_{s}}.$$

### 2.2. Noninverting Buck-Boost Converter (BB)

The noninverting buck-boost converter topology is shown in Figure 2, this power stage is composed of four switches and a single inductor *L*, and can operate in three modes [44]. For buck mode, switches $Q_{1}$ and $Q_{2}$ are turned in complementary manner, while $Q_{4}$ is turned off and $Q_{3}$ is turned on. For boost mode, switches $Q_{3}$ and $Q_{4}$ are turned on alternatively, while $Q_{1}$ is turned on and $Q_{2}$ is turned off. For buck-boost mode, the four switchers are operated as two high frequency interleaved half bridges. The mathematical model of the noninverting buck-boost converter is:
(8)$$\frac{di_{g}\left(t\right)}{dt}=\frac{V_{g}\;u_{1H}-\left(1-u_{2L}\right)v_{o}}{L}$$
(9)$$\frac{dv_{o}\left(t\right)}{dt}=\frac{i_{g}\left(1-u_{2L}\right)}{C_{o}}-\frac{v_{o}}{R_{o}C_{o}}$$Table 1 presents the expressions for the peak-to-peak ripple signals of the noninverting buck-boost converter. *T* is the switching period, and $D_{1}$ and $D_{2}$ are the steady-state duty cycles in boost mode and buck mode, respectively. The maximum inductor current of the noninverting buck-boost is:
(10)$$i_{gmax}=\left\{\begin{array}{c} \frac{P}{V_{g}}+\frac{V_{g}\left(V_{o}-Vg\right)}{V_{o}}\frac{T}{2L}\;\mathrm{for}\mathrm{boost}\mathrm{mode}\\ \frac{P}{V_{o}}+\frac{V_{o}\left(V_{g}-Vo\right)}{V_{g}}\frac{T}{2L}\;\mathrm{for}\mathrm{buck}\mathrm{mode}. \end{array}\right.$$
where *P* is the rated power of the converter. The single inductor is selected with a power rating of 3.2 kW, and a ripple current of 2 A at an input voltage $v_{g}=200$ V and an output voltage $v_{o}=300$ V, taking into account Equation (Equation 10) and the current ripple of Table 1. With an input voltage range $V_{g}$ from 0 V to 400 V, and an output voltage range $V_{o}$ from 0 V to 400 V. The selected value for the inductor is *L* = 330.75 μH with a switching frequency of 100 kHz.

### 2.3. Versatile Buck-Boost Converter (VBB)

The buck-boost converter of Figure 3 has two half bridge MOSFETs and an $R_{d}C_{d}$ damping network connected in parallel with the intermediate capacitor *C*. The pair of coupled inductors has an unitary ideal turns ratio $N_{2}/N_{1}$, a coupling coefficient *k* = 0.5, a mutual inductance *M* = 135 μH, and equal values for the primary ($L_{1}$) and secondary ($L_{2}$) self-inductances, being $L=L_{1}=L_{2}=270$ μH. In this analysis, the use of the state-space averaging (SSA) method to model the converter leads to the following set of differential equations [45]:
(11)$$\frac{di_{g}\left(t\right)}{dt}=\frac{L\left(V_{g}-v_{c}\left(1-u_{1L}\right)\right)-M\left(v_{o}-v_{c}\;u_{2H}\right)}{L^{2}-M^{2}}$$
(12)$$\frac{di_{L}\left(t\right)}{dt}=\frac{M\left(V_{g}-v_{c}\left(1-u_{1L}\right)\right)-L\left(v_{o}-v_{c}\;u_{2H}\right)}{L^{2}-M^{2}}$$
(13)$$\frac{dv_{c}\left(t\right)}{dt}=\frac{1}{C}\left(-i_{L}\;u_{2H}+i_{g}\left(-u_{1L}+1\right)-\frac{1}{R_{d}}\left(v_{c}-v_{cd}\right)\right)$$
(14)$$\frac{dv_{cd}\left(t\right)}{dt}=\frac{v_{c}-v_{cd}}{C_{d}\;R_{d}}$$
(15)$$\frac{dv_{o}\left(t\right)}{dt}=\frac{i_{L}}{C_{o}}-\frac{v_{o}}{R_{o}\;C_{o}}$$The converter introduced in [46] for high-voltage applications has an input voltage $V_{g}$ range of 200–400 V, and an output voltage range $V_{o}$ from 0 V to 400 V. Experimental efficiencies reported in [46] demonstrate high values over 95% in all the operation range, with a maximum value of 98% when the input and output voltages of the converter are near. Table 1 presents the current and voltage ripple for the VBB converter. The selected parameters are listed in Table 2. The values of the mutual inductor and the self-inductance were selected based on Table 1 to obtain a current ripple $\Delta i_{g}=2$ A at a switching frequency of 100 kHz for boost mode ($V_{g}=200$ V and $V_{o}=300$ V), which represents the most critical mode. The method to select the value for the components *C_d_*, *C* and *R_d_* is presented in [46]. The selection is a tradeoff between the capacitor size and ensures adequate and robust damping of the internal dynamics; the expression corresponds to:
(16)$$R_{d}\approx 0.65\sqrt{\frac{M}{C}},\;C_{d}\geq 8C.$$

## 3. Power Losses Methods

### 3.1. Electro-Thermal Model

The electro-thermal model is realized in PLECS, using the heat sink components for the power device SCT2450KEC employed for the switches of the buck-boost converters. This device is characterized by the maximum drain–source voltage equal to 1200 V and a permissible drain current equal to 10 A. The conduction, turn-on and turn-off switching losses are obtained from its datasheet, these were defined as simulation parameters, as shown in Figure 4. These relations are also linearly interpolated by the software. The software also enables implementing the dependence of switching energy losses. Separately, the energy values of the ON and OFF losses are entered, both for the MOSFET with diode model [47]. In addition, the thermal tool supports the quick entry of the MOSFET transient thermal impedance from the datasheet characteristics. The parameters of the MOSFET transient thermal impedance are selected from the datasheet for the power device SCT2450KEC.

### 3.2. Power Inductor Losses Calculation

The inductor power loss information is often provided by the core manufacturer. The power loss of the inductor estimation is calculated from the inductor core ($P_{core}$) and winding loss ($P_{dcr}$), using the expression:
(17)$$P_{Lossinductor}=P_{core}+P_{dcr}.$$$P_{core}$ is generated by the changing magnetic flux field within a material. A general form of the core loss formula for core loss density (PL) is [48]:
(18)$$PL=aB_{pk}^{b}f_{s}^{c}.$$
where a,b,c are constants determined from curve fitting, and $B_{pk}$ is defined as half of the AC flux swing:
(19)$$B_{pk}=\frac{B_{ACmx}-B_{ACmin}}{2}.$$The units are: mW/cm^3^ for PL; Tesla (T) for $B_{pk}^{b}$ and kHz for $f_{s}$. $B_{pk}$ can be obtained from the DC magnetization curve as follows:(20)$$B\left(H\right)={\left|\frac{a+bH+cH^{2}}{1+dH+eH}\right|}^{x}.$$
where a,b,c,d,e, and *x* are the constant parameters to fit the measured B-H curve data. *H* is the magnetic field intensity and is given by
(21)$$H_{ACmax}=\frac{Ni_{Lmax}}{l_{e}}$$
(22)$$H_{ACmin}=\frac{Ni_{Lmin}}{l_{e}}$$
where $i_{Lmax}$ and $i_{Lmin}$ are the maximum and minimum current points, respectively. *N* is the winding number of turns and $l_{e}$ is the magnetic path length. The DC resistance of a conductor is given by [49]:
(23)$$R_{wDC}=\rho_{w}\frac{l_{w}}{A_{w}}$$
where $l_{w}$ is the length of the conductor, its uniform cross-sectional area is $A_{w}$, and its resistivity $\rho_{w}$. Therefore, the wire loss caused by DC resistance is:(24)$$P_{dcr}=I_{rms}^{2}R_{wDC}$$$I_{rms}$ being the rms value of the ripple current applied to the inductor.

## 4. Voltage-Sensorless Predictive Controller for a Single Phase AC-DC Converter

The control in the AC-DC stage guarantees a high power factor, synchronizing the input current waveform with the electrical grid voltage. The signals needed to control the AC-DC stage include the line voltage $V_{ac}$, the input voltage $v_{g}$, the input current $i_{g}$ and the output voltage $v_{o}$. The control scheme in Figure 5 is proposed, where the voltage-control loop is implemented to regulate the output voltage $v_{o}$ and the current-control loop to regulate the input current $i_{g}$ amplitude and phase. A phase locked loop (PLL) is used to synchronize with the grid, and a normalized rectified sinusoidal reference obtained from the PLL is multiplied by the desired peak current value ($i_{peak}$) given by the voltage controller, to obtain the input current loop reference ($i_{ref}\left[n\right]$). Figure 5 reveals the proposed DC voltage sensorless control, which is based on computing the input voltage $V_{g}$ directly from the voltage line $V_{ac}$ by means of an absolute value function, obtaining the signal $v_{gest}\left[n\right]$. Furthermore, an online output voltage $v_{o}$ estimation is used to close the voltage loop using the computed duty cycles. The output voltage estimator of the SEPIC converter is calculated as follows:
(25)$$v_{oest}\left[n\right]=\frac{d[n]v_{g}[n]}{1-d[n]}.$$The following voltage output estimator is implemented for the BB and VBB converter:
(26)$$v_{oest}\left[n\right]=\frac{d_{2}[n]v_{g}[n]}{1-d_{1}[n]}.$$

Then, a proportional-integral digital control ($G_{vpi}\left(z\right)$) is used to obtain the reference peak current ($i_{peak}\left[n\right]$).

### 4.1. Proportional-Integral Voltage Controller

The proportional-integral controller regulates the output voltage of the DC bus to the reference value $V_{oref}$. The controller output is given by:(27)$$i_{peak}[n]=i_{Lp}[n]+i_{Li}[n].$$
where $i_{Lp}$ and $i_{Li}$ are the proportional and integral components, respectively. Their values are related in the following way:
(28)$$i_{Lp}[n]=K_{pv}\;e_{v}[n],\;i_{Li}[n]=K_{iv}\;T_{samp}\;e_{v}[n]+i_{Li}[n-1].$$
where $K_{pv}=2\pi \;f_{c}C_{o}$, $K_{iv}=K_{pv}/T_{i}$, $e_{v}\left[n\right]$ is the voltage error and $T_{samp}$ is the sampling period ($1/f_{samp}$). Hence, the bandwidth of the voltage loop depends on the proportional coefficient ($K_{pv}$). The value of the crossover frequency (CF) for the voltage loop ($f_{c}$) should be lower than the CF for the current loop. The location of the PI zero should be lower than $f_{c}$ ($1/\left(2\pi T_{i}\right)<f_{c}$) [50]. The zero corner frequency for the current loop should be much smaller than $f_{s}$ being 10 times below $f_{s}$ [51,52]. Therefore, the CF corresponds to 10 kHz for the current loop and 2500 Hz for the voltage loop. Finally, the location for the PI zero of the voltage loop is 250 Hz.

### 4.2. Predictive Digital Current Programmed Control

The predictive digital current programmed control (PDCC) has been presented in [53,54]. The aim of this strategy is to compute the duty cycle $d_{x}$ in the n+1 time-sampling interval. The analysis for the buck-boost converters presented for the small model allows to find the converter’s current output slope $\frac{di_{g}}{dt}$ for the SEPIC (see Figure 1), the noninverting buck-boost (see Figure 2) and versatile buck-boost (see Figure 3) converters. Current $i_{g}$ has a periodic triangular ripple waveform with a rising slope $m_{1}$ and a falling slope $-m_{2}$. Table 3 presents the converter current $i_{g}$ ripple waveform slopes based on the equation for $\frac{di_{g}}{dt}$ from (Equation 1), (Equation 8) and (12) for the boost and buck modes.

The Euler approximation leads to the following discrete-time output current expression, assuming the converter’s current output slope $\frac{di_{g}}{dt}\approx \frac{i_{g}\left[n+1\right]-i_{g}\left[n\right]}{T}$ from the averaged model.
(29)$$i_{g}\left[n+1\right]=i_{g}\left[n\right]+T\left(m_{1}+m_{2}\right)d_{x}\left[n\right]-m_{2}T.$$

Hence, the resulting expression of the duty cycle is:(30)$$d_{x}\left[n+1\right]=-d_{x}\left[n\right]+\frac{1}{(m_{1}+m_{2})T}e_{i}\left[n\right]+2\frac{m_{2}}{m_{1}+m_{2}},$$
where *x* corresponds to the operating mode of the bidirectional BB and VBB converters (*x* = 1 for boost mode, *x* = 2 for buck mode). In the case of the SEPIC converter $d_{x}\left[n\right]=d\left[n\right]$. Using the expressions of the output current slopes, $m_{1}$ and $-m_{2}$, in Table 3 at (Equation 30), the expression of $m_{1}+m_{2}$ for the SEPIC converter is
(31)$$m_{1}+m_{2}=\frac{v_{c}+v_{o}}{L_{1}}.$$The corresponding sum of slopes for the BB converter can be expressed as: (32)$$m_{1}+m_{2}=\left\{\begin{array}{c} \frac{v_{o}}{L}\;\mathrm{for}\mathrm{boost}\mathrm{mode}\\ \frac{v_{g}}{L}\;\mathrm{for}\mathrm{buck}\mathrm{mode}. \end{array}\right.$$Furthermore, for the VBB, it can be seen that
(33)$$m_{1}+m_{2}=\left\{\begin{array}{c} \frac{L\;v_{c}}{L^{2}-M^{2}}\;\mathrm{for}\mathrm{boost}\mathrm{mode}\\ \frac{M\;v_{c}}{L^{2}-M^{2}}\;\mathrm{for}\mathrm{buck}\mathrm{mode}. \end{array}\right.$$The expression $m_{2}/\left(m_{1}+m_{2}\right)$ for the SEPIC converter is:
(34)$$\frac{m_{2}}{m_{1}+m_{2}}=\frac{v_{c}+v_{o}-V_{g}}{v_{c}+v_{o}}.$$For the VBB it is given by:
(35)$$\frac{m_{2}}{m_{1}+m_{2}}=\left\{\begin{array}{c} \frac{-L\left(V_{g}-v_{c}\right)+M\left(v_{o}-v_{c}\right)}{L\;v_{c}}\;\mathrm{for}\mathrm{boost}\mathrm{mode}\\ \frac{-L\left(V_{g}-v_{c}\right)+Mv_{o}}{M\;v_{c}}\;\mathrm{for}\mathrm{buck}\mathrm{mode}. \end{array}\right.$$Furthermore, for the BB converter can be expressed as:(36)$$\frac{m_{2}}{m_{1}+m_{2}}=\left\{\begin{array}{c} \frac{V_{g}+v_{o}}{V_{g}}\;\mathrm{for}\mathrm{boost}\mathrm{mode}\\ \frac{v_{o}}{V_{g}}\mathrm{for}\mathrm{buck}\mathrm{mode}. \end{array}\right.$$The Equations (Equation 33) and (Equation 35) for the BBV converter can be simplified by substituting the voltage of the intermediate capacitor $v_{c}$ by $V_{g}$ in buck mode operation and by $V_{o}$ in boost mode operation. Furthermore, for the SEPIC converter by substituting $v_{c}$ by $V_{g}$ in (Equation 31) and (Equation 34).

## 5. Simulation and Real-Time HIL Results

In this section, power conversion efficiency results of the VBB, BB and SEPIC converters are compared. Once the high efficiency of the VBB converter is verified, the experimental, simulation and HIL (Hardware in the loop) tests are presented to validate the proposed DC voltage sensorless control using the VBB converter.

### 5.1. Efficiency Results

In this section, efficiency results of the buck-boost converter topologies for AC-DC application are presented. The design of the buck-boost converters are realized under the same characteristic of switching frequency and peak to peak current ripple presented in Section 2. The thermal model to obtain the conduction and switches losses for the MOSFET was implemented in PLECS, this simulation is carried out using the heat sink components for the power device SCT2450KEC employed for the DC-DC converters. The power inductor losses estimation for the inductors takes into account the inductor design presented in [46], where the core used is the 77908 from Magnetics, which has a Kool Mμ material with a core relative permeability coefficient of 26, and the windings wire size is 18 AWG. The generalized block diagram of the buck-boost converters in AC–DC applications for current regulation is shown in Figure 6. In this case, to achieve a high power factor, a normalized rectified sinusoidal reference is synchronized to the line input voltage ($V_{line}$) and multiplied by the desired peak current value ($i_{peak}$) to obtain the current loop reference ($i_{ref}\left[n\right]$). The PDCC loop control described in Section 2 is used to track the rectified sinusoidal current reference. Figure 7a shows that the converter output voltage ($V_{o}$) is fixed to 200 V, which is lower than the peak value of the input voltage ($v_{g}$) that corresponds to the rectified voltage $V_{line}$, ensuring operation in both boost and buck modes at each semi–period of the grid. Figure 7b shows the results for the current control where the waveforms are $i_{g}$ and its RMS current value, this last is about 4 A. The controlled current ripple has a peak-to-peak amplitude of about 2 A for an input voltage of 300 V and an output voltage of 200 V. The corresponding peak current reference is $i_{peak}=6$ A. The simulated results show a good current reference tracking under changes of the operation mode (boost or buck) and peak current reference ($i_{peak}$), validating in this way the good control performance. The efficiencies of the converters are calculated taking into account the switches losses, the inductors core losses, and the damping resistor losses $R_{d}$ for the versatile buck-boost converter, as follows:(37)$$\eta =1-\frac{P_{loss}}{P_{in}}$$

Being $P_{loss}$ the power losses for the inductor, switches and resistor elements of the converters. $P_{in}$ is the input power, whose rms value is 900 W. Figure 7c shows the efficiency results in the time domain over one period of the grid signal (20 ms), where the efficiency changes for different operating points, while the operation modes can be determined by the input and output voltages seen in Figure 7a. For all converters, the highest efficiency is reached when the input current and voltage pass through the zero crossings, that is, when the converters do not process power. The SEPIC and versatile buck-boost converters show a similar efficiency behaviour when the output voltage is much higher than the input voltage for boost mode. The VBB converter has a better power conversion efficiency, higher than 95% for all the operation range while the SEPIC converter has a minimum efficiency near to 88% and the noninverting BB above 93%. In addition, the output currents $i_{o}$ of all buck-boost converters are shown in Figure 7d. These results show that the versatile buck-boost converter has the lowest output current ripple. This advantage is very relevant since it reduces the output filter requirements in the converter output for many applications.

### 5.2. Experimental Results

The HIL test has been implemented into two subsystems: the plant and the controller. The plant subsystem corresponding to the AC-DC stages is deployed on the PLECS RT Box 1 for Hardware-in-the-loop (HIL) testing. The controller subsystem has been flashed to a low-cost digital signal controller (DSC), which corresponds to the Texas Instruments LAUNCHXL-F28069M as can be seen in Figure 8. The controller implements a double loop control algorithm to regulate the AC-DC converter’s output voltage. The sampling time of the plant is 2 μs.

### 5.3. Inner Loop Current Control Results

A dynamic response test of the current control has been performed to evaluate the inner loop based on a predictive digital strategy. Figure 9 shows the experimental results for an input rms voltage of 220 V and an input line voltage frequency ($f_{i}$) of 50 Hz. The signals sampled for the control are $v_{g}$, $v_{o}$, $v_{ac}$ and $i_{L}$. The sampling time is 40 μs. The control scheme for the inner loop is shown in Figure 6. Experimental results in Figure 9 show that the input current ($i_{g}$) of the VBB converter is in phase with the input voltage ($v_{g}$) while the desired peak current ($i_{peak}$) changes from 4 A to 8 A. The converter output voltage ($v_{o}$) is fixed to 200 V, which is lower than the peak value of the input voltage ($v_{g}$) that corresponds to the rectified voltage $v_{ac}$, ensuring operation in both boost and buck modes in each grid half-period. The experimental results show good current reference tracking under changes of the operation mode (boost or buck) and peak current reference ($i_{peak}$). The inner loop control depends on the inductor parameter of the model, this coupled inductor value depends on the temperature, DC current and switching frequency [55,56]. Therefore, a sensitivity analysis for the current control of each converter parameter operating in boost mode ($V_{g}=$ 200 V and $V_{o}=$ 400 V) and buck mode ($V_{g}=$ 400 V and $V_{o}=$ 200 V) both with a $i_{gref}=$6 A, is shown in Figure 10. To reproduce the parameter mismatch, parameters in the model (*L*, *M*, *C*, *C_d_* and *R_d_*) are varied for ±20%, in order to investigate the individual effect of each parameter in the performance of the $MAPE(i_{g})$, which is defined as
(38)$$MAPE(i_{ref})=\frac{100\%}{n}\cdot \sum_{t=1}^{n}\left|\frac{i_{ref}-i_{g}}{i_{ref}}\right|,$$
where *n* is the number of steps of the simulation. This measure corresponds to the mean absolute percentage error between the input current *i_g_* and its respective reference *i_ref_*. Therefore, from the obtained results of Figure 10, it can be stated that *L* and *M* have a higher sensibility in the input current tracking error for boost mode when the mutual and self inductance tolerance have values around ±20%. However, even in the worst case the $MAPE(i_{g})$ has a small variation of 1.8% in the boost mode. On the other hand, it is evident that the deturning of the other variables of the converter (*C*, *C_d_* and *R_d_*) do not have a relevant effect on the $MAPE(i_{g})$.

### 5.4. DC Voltage Sensorless Control Results

For the results shown in Figure 11, a 50 Ω load resistor has been used as is shown in Figure 5, the output capacitor is *C_o_* = 3 mF and the crossover frequency of the voltage loop is 1.5 kHz. In Figure 11, the experimental results are presented for double loop control operating with sensor and sensorless modes. Until the first second the control strategy is in sensor mode, meaning that all the control variables are sensed. After the first second, the control strategy switches from sensor to sensorless mode, meaning that both *v_g_* and *v_o_* are estimated. The control strategy has a good output voltage estimation during the operation into the sensorless mode with a small relative error, as shown in Figure 11a. In sensorless mode, the only sensed variables are grid voltage *V_ac_* and current *i_g_*. Therefore, the proposed sensorless strategy allows reducing a critical sensor as the output voltage case reduces the converter costs, increases the power density, and increases the converter reliability without increasing the computational cost in the control strategy. Figure 11 shows a good performance of the control strategy in both sensor and sensorless mode, which validates the excellent estimation performed by the control algorithm. As can be observed at Figure 11, the output voltage is well regulated at 200 V for the DC voltage sensorless control, with a current input peak of 6 A. Figure 12 show experimental and simulated results for variations of the output voltage reference from 220 V to 200 V with a 100 Ω resistor load. The input current decreases following the changes of the output voltage, demonstrating the good performance of the DC voltage sensorless control for large variations of the voltage reference. A step variation in the load is shown in Figure 13, where the load current changes from 2 A to 4 A, and then from 4 A to 2 A. The voltage controller regulates *v_o_* at the desired value *v_oref_* = 200 V when the resisistive load changes between 100 Ω and 50 Ω. The figures also demonstrate a good agreement between the experimental and simulation results, which validates the adequate operation of the proposed control method.

## 6. Conclusions

This paper presents a DC voltage sensorless predictive control for the versatile buck-boost converter operating as PFC single-phase rectifier. The SEPIC, BB and VBB converters were compared with respect to power conversion efficiency. For a fair comparison, common power stage design parameters were considered for all buck-boost converters and the same discrete predictive current control was applied to them. A power inductor losses method was used to calculate the efficiency, and a PLECS thermal simulation has been developed to estimate the the MOSFET and inductor losses. Simulated results confirm the advantages of the proposed versatile buck-boost converter operating as PFC single-phase rectifier among which stands out high power conversion efficiency over a wide operation range. In addition, real-time HIL tests for the VBB converter have been performed using a PLEC RT BOX1 device. A double loop sensorless control to regulate the output voltage was implemented in a DSC Texas Instruments LAUNCHXL-F28069M. Temporal and thermal simulations and real-time HIL results verify the correctness of the proposed control for the versatile buck-boost converter operating in an AC-DC application.

## Figures and Tables

**Figure 1 sensors-21-05107-f001:**
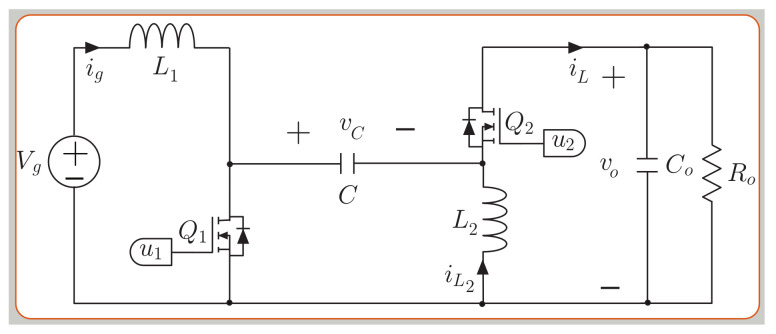
Topology of the SEPIC converter.

**Figure 2 sensors-21-05107-f002:**
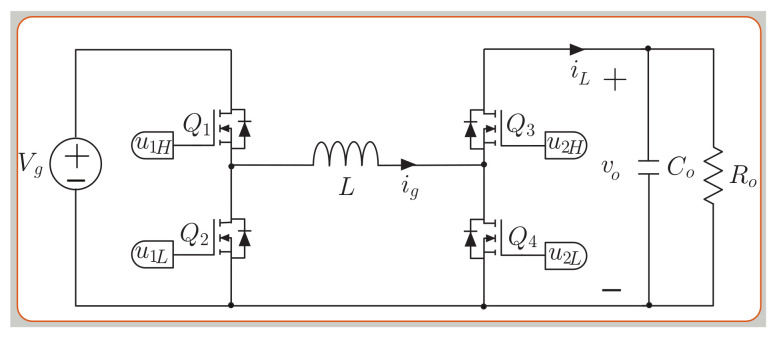
Topology of the noninverting buck–boost converter.

**Figure 3 sensors-21-05107-f003:**
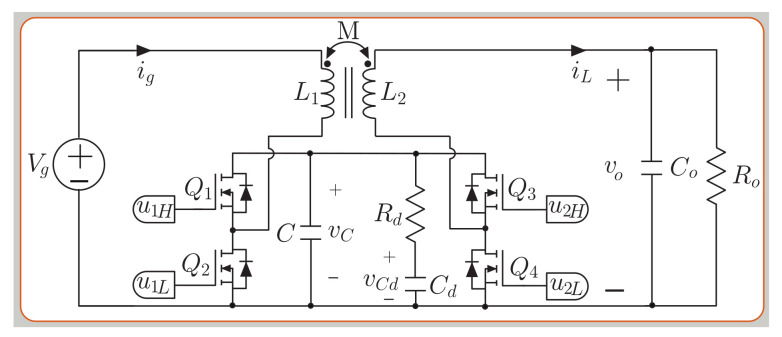
Topology of the buck–boost versatile converter, VBB.

**Figure 4 sensors-21-05107-f004:**
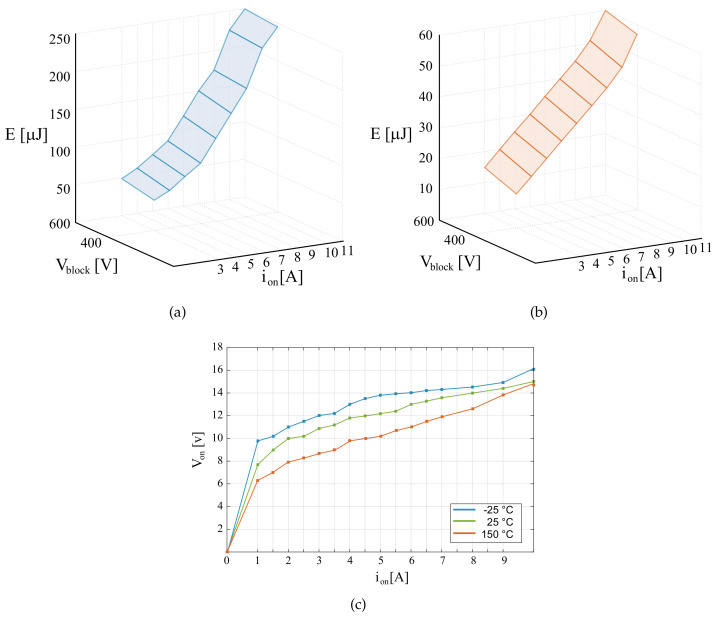
Switching and conduction losses of SCT2450KEC parameters of PLECS: (**a**) Turn on losses, (**b**) Turn off losses (**c**) Conduction losses.

**Figure 5 sensors-21-05107-f005:**
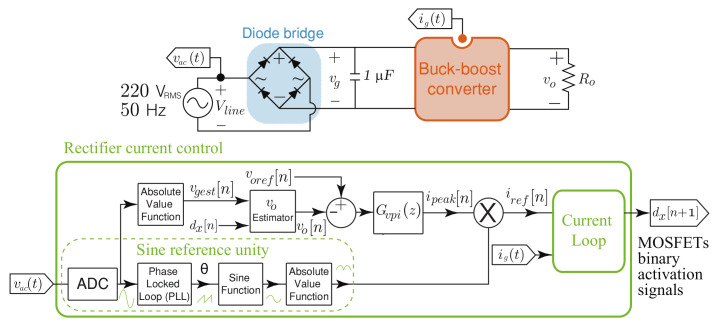
Block diagram of the buck-boost converter in an AC-DC application where its output voltage is regulated.

**Figure 6 sensors-21-05107-f006:**
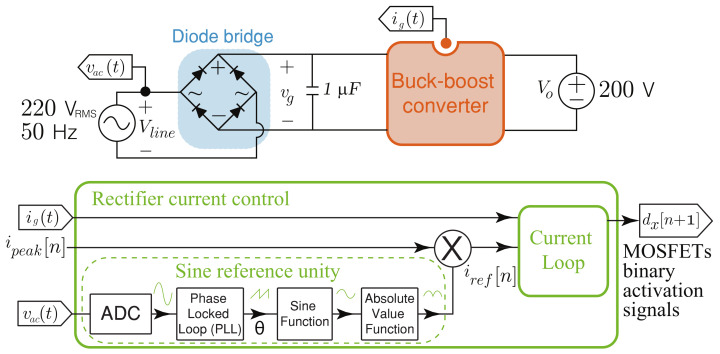
Block diagram of the buck-boost converter in an AC-DC application where its input current is regulated.

**Figure 7 sensors-21-05107-f007:**
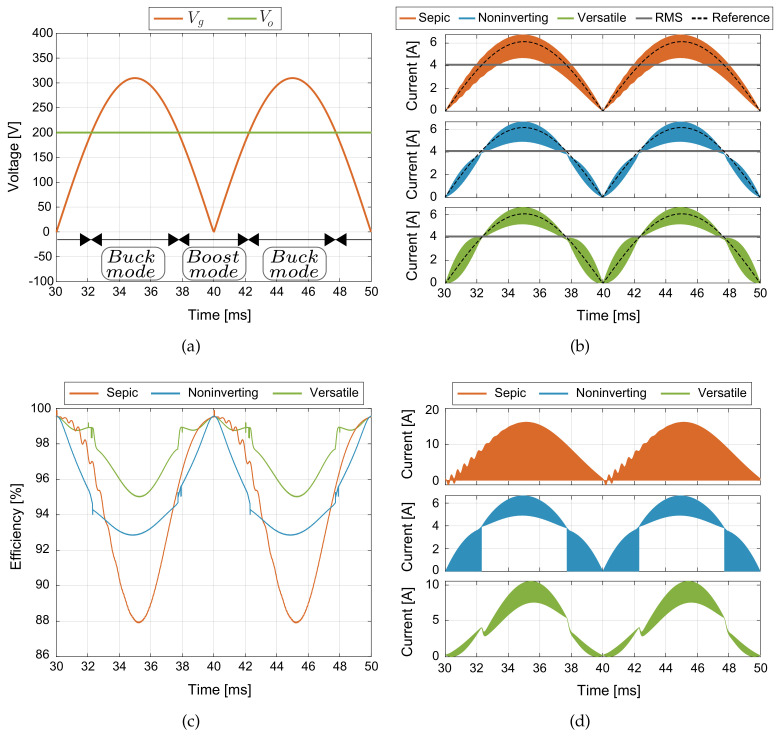
PLECS thermal simulation: (**a**) input and output voltage waveforms, (**b**) current control waveforms, (**c**) AC-DC conversion efficiency results over one period of the grid signal (20 ms), (**d**) output current converter.

**Figure 8 sensors-21-05107-f008:**
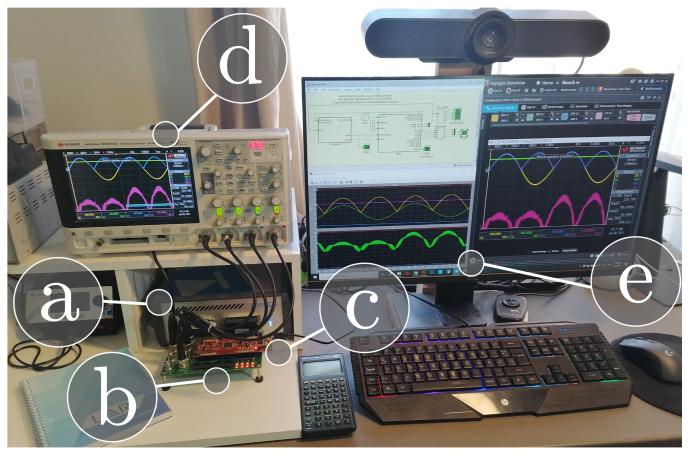
Hardware in-the-loop experimental setup: (**a**) PLECS RT-box , (**b**) RT Box LaunchPad Interface, (**c**) Texas Instruments LAUNCHXL-F28069M, (**d**) oscilloscope, (**e**) computer to program the RT Box and the microcontroller, perform the simulations, and communicate the data with the oscilloscope.

**Figure 9 sensors-21-05107-f009:**
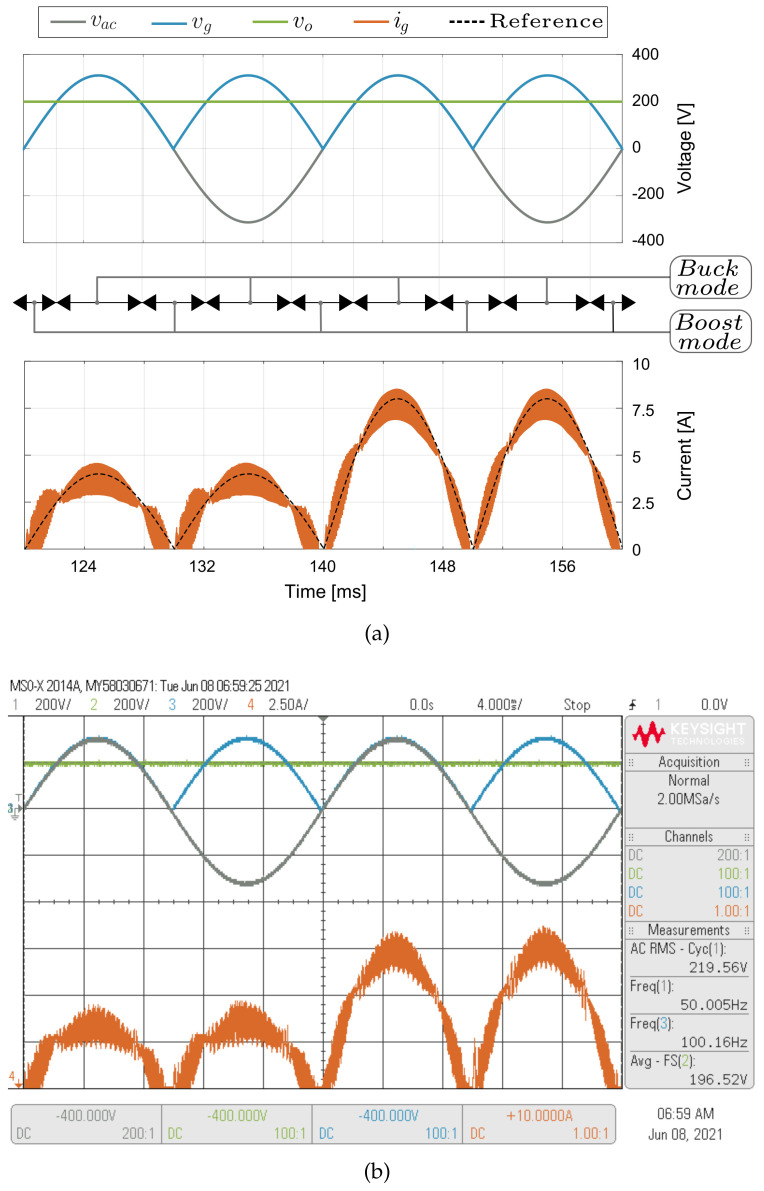
Simulated (**a**) and experimental (**b**) dynamic behavior of the predictive digital current input control when the reference *i_ref_* changes from 4 A to 8 A with a fixed output voltage *V_o_* = 200 V. CH1: *v_ac_* (200 V/div), CH2: *v_o_* (200 V/div), CH3: *v_g_* (200 V/div), CH4: *i_g_* (2.5 A/div) and a time base of 4 ms.

**Figure 10 sensors-21-05107-f010:**
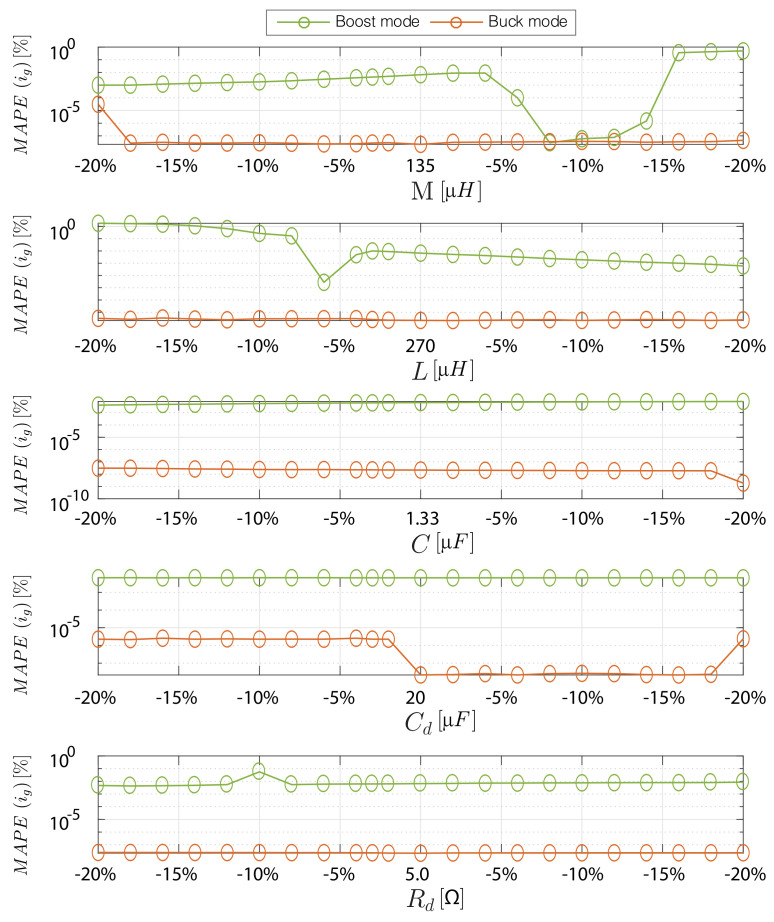
Sensitivity analysis of the predictive inner current loop under converter parameters variations ( *M*, *L*, *C*, *C_d_* and *R_d_*) operating in boost mode (*V_g_* = 200 V and *V_o_* = 400 V) and buck mode (*V_g_* = 400 V and *V_o_* = 200 V) both with a *i_ref_* = 6 A.

**Figure 11 sensors-21-05107-f011:**
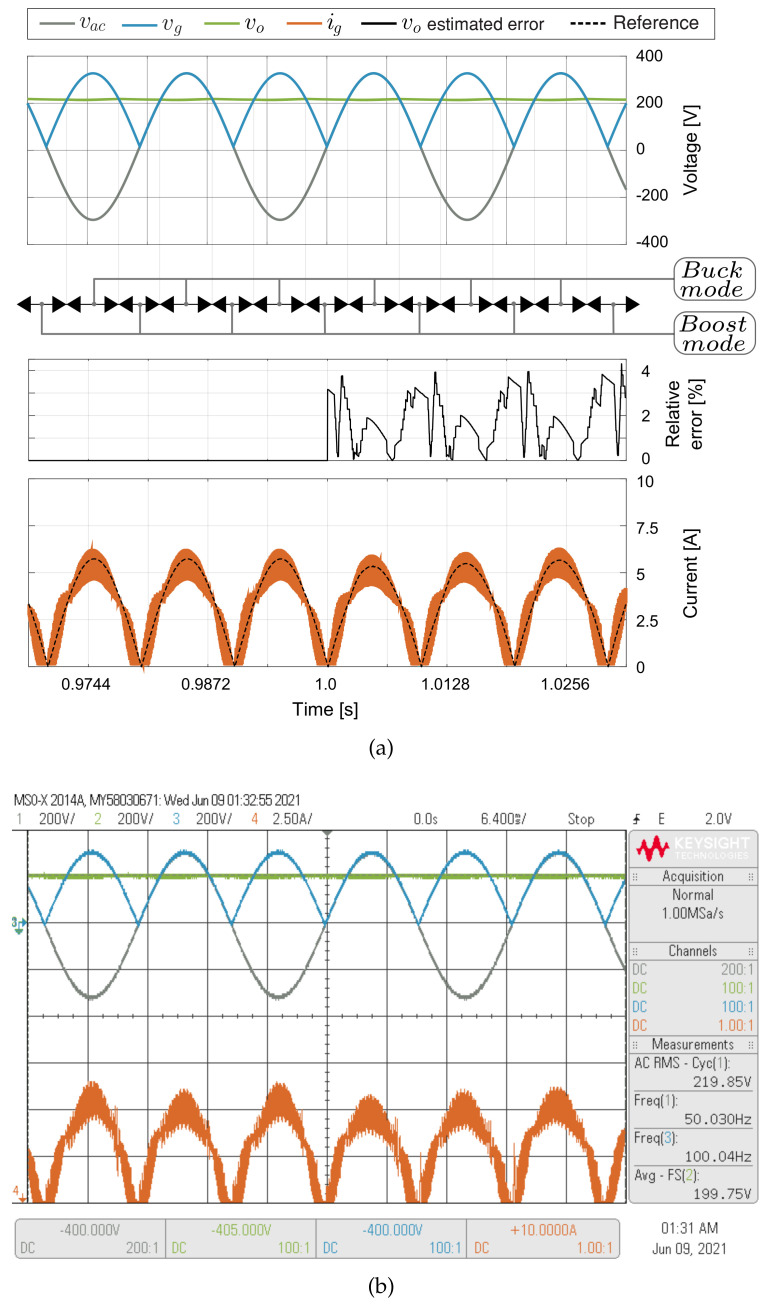
Simulated (**a**) and experimental (**b**) response of the double loop when the voltage sensorless control is switched on at 1 s with *v_oref_* = 200 V. CH1: *v_ac_* (200 V/div), CH2: *v_o_* (200 V/div), CH3: *v_g_* (200 V/div), CH4: *i_o_* (2.5 A/div) and a time base of 6.4 ms.

**Figure 12 sensors-21-05107-f012:**
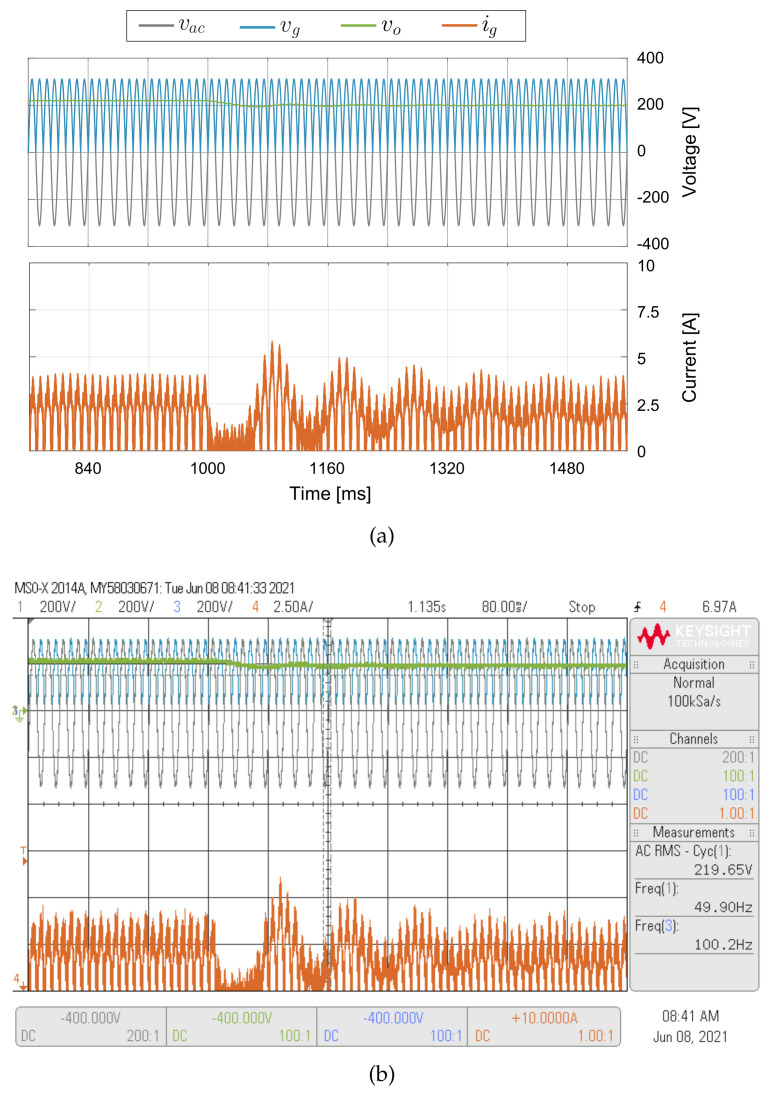
Simulated (**a**) and experimental (**b**) dynamic behavior of the double loop when the reference *v_oref_* changes with steps of 20 V from 220 V to 200 V. CH1: *v_ac_* (200 V/div), CH2: *v_o_* (200 V/div), CH3: *v_g_* (200 V/div), CH4: *i_g_* (2.5 A/div) and a time base of 80 ms.

**Figure 13 sensors-21-05107-f013:**
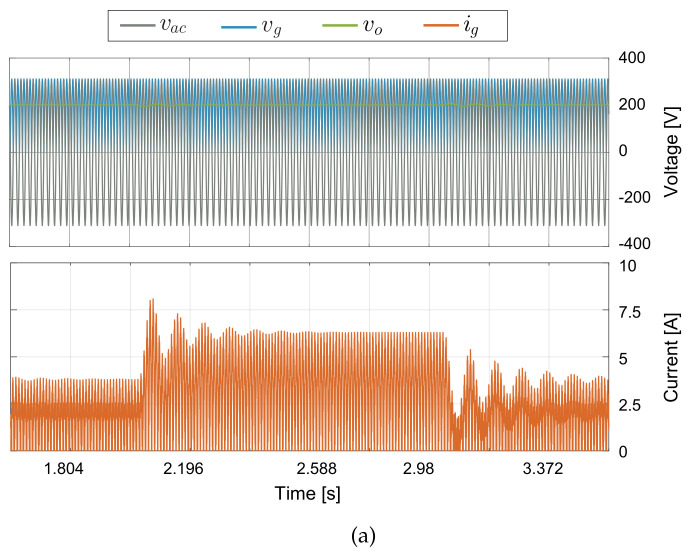
Simulated (**a**) and experimental (**b**) dynamic behavior of the double loop for a 50% step change in the load. CH1: *v_ac_* (200 V/div), CH2: *v_o_* (200 V/div), CH3: *v_g_* (200 V/div), CH4: *i_g_* (2.5 A/div) and a time base of 196 ms.

**Table 1 sensors-21-05107-t001:** Peak-to-peak ripple amplitudes for BB and VBB.

BB Converter		Buck Mode		Boost Mode
$\Delta i_{g}$		$\frac{\left(V_{g}-V_{o}\right)V_{o}T}{V_{g}L}$		$\frac{V_{g}\left(V_{o}-V_{g}\right)T}{V_{o}L}$
**VBB Converter**		**Buck Mode**		**Boost Mode**
$\Delta i_{L}$		$\frac{V_{o}T\left(V_{g}-V_{o}\right)L}{V_{g}\left(L^{2}-M^{2}\right)}$		$\frac{V_{g}T\left(V_{o}-V_{g}\right)M}{V_{o}\left(L^{2}-M^{2}\right)}$
$\Delta i_{g}$		$\frac{V_{o}T\left(V_{g}-V_{o}\right)M}{V_{g}\left(L^{2}-M^{2}\right)}$		$\frac{V_{g}T\left(V_{o}-V_{g}\right)L}{V_{o}\left(L^{2}-M^{2}\right)}$
$\Delta v_{c}$		$\frac{D_{2}I_{L}}{C}T\left(D_{2}-1\right)$		$\frac{D_{1}I_{L}}{C}T$

**Table 2 sensors-21-05107-t002:** Selected components and parameters of the versatile buck-boost converter.

Parameter	Value or Type
Input voltage *V_g_*	0–400 V
Output voltage *V_o_*	100–400 V
Rated Power	3.2 kW
Switching frequency *f_s_*	100 kHz
Output capacitor *C_o_*	28 μF
Damping capacitor *C_d_*	20 μF
Intermediate capacitor *C*	1.32 μF
Coupled inductor	*M* = 135 μH and *L* = 270 μH
Damping resistance *R_d_*	5 Ω

**Table 3 sensors-21-05107-t003:** Slopes of *i_g_* current waveform.

SEPIC	$m_{1}$	$-m_{2}$
converter	$\frac{V_{g}}{L_{1}}$	$\frac{-v_{c}-v_{o}+V_{g}}{L_{1}}$
**BB converter**	$m_{1}$	$-m_{2}$
Buck	$\frac{V_{g}-v_{o}}{L}$	$-\frac{v_{o}}{L}$
Boost	$\frac{V_{g}}{L}$	$\frac{V_{g}-v_{o}}{L}$
**VBB converter**	$m_{1}$	$-m_{2}$
Buck	$\frac{L\left(V_{g}-v_{c}\right)-M\left(v_{o}-v_{c}\right)}{L^{2}-M^{2}}$	$\frac{L\left(V_{g}-v_{c}\right)-Mv_{o}}{L^{2}-M^{2}}$
Boost	$\frac{LV_{g}-M\left(v_{o}-v_{c}\right)}{L^{2}-M^{2}}$	$\frac{L\left(V_{g}-v_{c}\right)-M\left(v_{o}-v_{c}\right)}{L^{2}-M^{2}}$

## Data Availability

Not applicable.

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
