# Peer review of "DC Voltage Sensorless Predictive Control of a High-Efficiency PFC Single-Phase Rectifier Based on the Versatile Buck-Boost Converter"

_sensors, 2021, doi:10.3390/s21155107_

Round 1

Reviewer 1 Report

-Lines 72-74. You claim that the calculation of switching and conduction losses of the switches with the inductor losses represents an 'excellent' approach to estimate the converter's efficiency. Instead of saying that it is an excellent approach you should justify with references such statement.

-Equations 2-4. 'I2' is not coherently defined, as it also appears as 'i2'. Please, choose one notation or the other but do not mix them. Besides, neither i1 nor i2 have been pinpointed in Fig. 1, and there are signals in Fig1, that are not present in equations 1-4. Similar issues are found in equations 8-9 and Fig.2 and beyond. Please, make the equations coherent with the schemes.

-Table 2. You should justify the value for each variable.

-Table 3. It is really difficult for the reader to guess what each column and row of this table means. Please, clarify this and if not, remove the table.

-Improve the quality of Fig.5, especially for the upper scheme.

-Lines 161-163. Justify the statements you made for the selection of crossover frequency for the voltage loop and for the current loop. Justify the statement for the location of PI zero.

-Table 4 and equations 31, 33 and 35. According to the info provided in Table 4, there are errors either in Table 4 or in eqs 31, 33, 35. Please, check the errors in the analytical expressions m1+m2 and

m2/(m1+m2)for several converters. And use a common notation for variables Vc and Vac, as they appear as vc and vac also.

-Lines 173-175. The obtention of equation 29 has not been appropiately explained in Lines 173-175. Please, clarify better how to obtain equation 29. And such clarification should appear inmediately after or before eq 29, and not at the end of the section.

- Between lines 182-183. Define the acronym called 'PACC loop control', as its definition does not appear in the text. Regarding the sentence 'the simulated results show a good current reference tracking under changes of operation mode and peak current reference, validating in this way the good control performance', you cannot claim this unless you confront the current signals with their control references in the same graph and such claim can be easily noticed by the reader. And you haven't done this. Please present the current signals with their current control references in order to claim that.

-Lines 188-190. Pinpoint this area in the corresponding graph.

-Fig 8 and Fig 10 should be more related.

-Lines 209-211. Cite the figure where this happens.

-Lies 214-216. Again, you cannot claim this unless you confront current signals with current reference signals in the same graph and you should refer to such graph. Besides, you should clarify in the graph when it works in buck and boost modes.

-Fig 11. Bad labeling of several components.

-Lines 225-227. Not enough reasons provided to make such statement.

-To finish with, I do not see clearly the novelty of this work, since in those figures where you switch from a sensor to a sensorless schemes, no evident improvements can be observed. Besides, you claim in the introduction that the use of sensorless schemes reduce the coupling noise and i do not see that noise reduction in those mentioned graphs.

Author Response

Reviewer#1, Concern # 1: Lines 72-74. You claim that the calculation of switching and conduction losses of the switches with the inductor losses represents an 'excellent' approach to estimate the converter's efficiency. Instead of saying that it is an excellent approach you should justify with references such statement.

Author response: Thank you very much for the comment. The reviewer is right. We have changed the wording of the paragraph and added references that justify the losses studied in our analysis.

Reviewer#1, Concern # 2: Equations 2-4. 'I2' is not coherently defined, as it also appears as 'i2'. Please, choose one notation or the other but do not mix them. Besides, neither i1 nor i2 have been pinpointed in Fig. 1, and there are signals in Fig1, that are not present in equations 1-4. Similar issues are found in equations 8-9 and Fig.2 and beyond. Please, make the equations coherent with the schemes.

Author response: Thank you very much for noticing. All the mentioned equations have already been corrected in the new version of the article.

Reviewer#1, Concern # 3: Table 2. You should justify the value for each variable.

Author response:  Thank you for the observation. We included a description of the procedure to select the converter components listed in Table2.

Reviewer#1, Concern # 4: Table 3. It is really difficult for the reader to guess what each column and row of this table means. Please, clarify this and if not, remove the table

Author response:  Thank you for your suggestion, we removed Table 3 and added in the description that the transient thermal impedance is taken from the device datasheet.

Reviewer#1, Concern # 5: Improve the quality of Fig.5, especially for the upper scheme.

Author response:  Thank you very much for your comment. We have improved all the figures to enhance their viewing and understanding.

Reviewer#1, Concern # 6: Lines 161-163. Justify the statements you made for the selection of crossover frequency for the voltage loop and for the current loop. Justify the statement for the location of PI zero.

Author response: Thank you for your recommendation. We included new references to justify the statements and added information about the selection of crossover frequency.

Reviewer#1, Concern # 7: Table 4 and equations 31, 33 and 35. According to the info provided in Table 4, there are errors either in Table 4 or in eqs 31, 33, 35. Please, check the errors in the analytical expressions m1+m2 and m2/(m1+m2) for several converters. And use a common notation for variables Vc and Vac, as they appear as vc and vac also

Author response: Thank you very much for noticing. All the mentioned equations and Table 4 have already been corrected in the new version of the article.

Reviewer#1, Concern # 8: -Lines 173-175. The obtention of equation 29 has not been appropiately explained in Lines 173-175. Please, clarify better how to obtain equation 29. And such clarification should appear inmediately after or before eq 29, and not at the end of the section.

Author response: Thank you very much for the comment. The reviewer is right. We have changed the equations in the text, which corresponds in the new article version to equations (33) and (35) for the BBV converter.

Reviewer#1, Concern # 9: Between lines 182-183. Define the acronym called 'PACC loop control', as its definition does not appear in the text. Regarding the sentence 'the simulated results show a good current reference tracking under changes of operation mode and peak current reference, validating in this way the good control performance', you cannot claim this unless you confront the current signals with their control references in the same graph and such claim can be easily noticed by the reader. And you haven't done this. Please present the current signals with their current control references in order to claim that.

Author response: Thank you very much for the comment, we changed the acronym to PDCC and defined it in Subsection 4.2. We grouped Figures from 7 to 10 in Figure 7 (new paper version) and added the corresponding current reference signal. 

Reviewer#1, Concern # 10: Lines 188-190. Pinpoint this area in the corresponding graph.

Author response: Thank you for your recommendation. We changed the text and highlighted the buck and boost mode in the voltage Figures 7, 9 and 11.

Reviewer#1, Concern # 11: Fig 8 and Fig 10 should be more related.

Author response: Thank you very much for the comment. We grouped Figures from 7 to 10 in Figure 7 (new paper version).

Reviewer#1, Concern # 12: -Lines 209-211. Cite the figure where this happens.

Author response: Thank you very much for the comment. We added the corresponding cite to the figure.

Reviewer#1, Concern # 13: - -Lies 214-216. Again, you cannot claim this unless you confront current signals with current reference signals in the same graph and you should refer to such graph. Besides, you should clarify in the graph when it works in buck and boost modes.

Author response: Thank you for your recommendation. We highlighted the mode operation in the figure and added the reference current for the simulated results (see Figure 11 in new paper version).

Reviewer#1, Concern # 14: - -Fig 11. Bad labeling of several components.

Author response: Thank you very much for noticing.  We corrected the labeling for the setup figure (Figure 8 in new version paper).

 Reviewer#1, Concern # 15: -Lines 225-227. Not enough reasons provided to make such statement.

Author response: The authors agree with the reviewer. We have included in Figure 11 (new paper version) the relative error of the estimation of the output voltage. The small relative error obtained during the control in the sensorless mode, demonstrates the excellent performance of the proposed estimator. In addition, the fact that the control operating in sensor and sensorless mode is equivalent, as shown in Figure 11, also validates that the sensorless mode works correctly.

Reviewer#1, Concern # 16: -To finish with, I do not see clearly the novelty of this work, since in those figures where you switch from a sensor to a sensorless schemes, no evident improvements can be observed. Besides, you claim in the introduction that the use of sensorless schemes reduce the coupling noise and i do not see that noise reduction in those mentioned graphs.

Author response: Thank you for raising this concern. First, the voltage sensing is based on the AD8031 high-speed amplifier, which has low distortion and fast settling time, making it ideal as buffers to single-supply ADCs. Given the selection of such a good amplifier for sensing, a significant variation should not be observed concerning noise. However, being able to design a control strategy without having to require an additional sensor has many advantages. The reduction of space in the implementation, minimizing costs, and the eventual possibility of being susceptible to noise depend significantly on the quality of the sensing and its conditioning and wiring to the control. Second, the output voltage sensor is considered one of the most critical components of a rectifier converter. In this sense, high reliability is obtained by reducing the number of sensors, especially if it comes to this critical sensor. Finally, according to your previous answer, showing a similar behavior between the control operating between the sensor and sensorless modes has many implicit advantages, among which stand out the reduced system cost, increased reliability without increasing the computational cost or complexity. We have included these advantages in the article to highlight the differences between the sensor and sensorless strategies.

Reviewer 2 Report

The article is properly written in terms of engineering, but the scientific value is low.

The reviewer has some comments and suggestions listed below:

In contributions  autors written that (line 75) "In this paper, a new single-phase PFC rectifier based on the versatile buck-boost converter is proposed. The solutions presented in article there are not new. (DOI:10.1109/ICIT.2015.7125236) an there are know. Please do not write that this is now solution.

At the beginning of the introduction, the authors describe sensorless solutions in the electric drive, but in no part of the text did they indicate that these solutions concern only the lack of a speed sensor or the position of the machine shaft. This is important because the term "sensorless" in electric drives only applies to systems that do not measure angle or speed.

The value of the current depends on the parameters of inductive elements, therefore the authors should note how the quality of the presented solution changes depending on the change in temperature and parameters of inductive elements. Pleas add some comments in text. (http://przyrbwn.icm.edu.pl/APP/PDF/133/app133z4p65.pdf)

Table 3 schould be push to appendix.

The sentence "Figure 7 shows that the converter output voltage ...." should be more precise and contain information about which version of the layout it refers to. The note also applies to the description of Figure 7.

Please explain more clear to the reader why the efficiency of the presented systems changes over time in the presented figure 9. Please add a comment.

Author Response

Reviewer#2, Concern # 1: In contributions  autors written that (line 75) "In this paper, a new single-phase PFC rectifier based on the versatile buck-boost converter is proposed. The solutions presented in article there are not new. (DOI:10.1109/ICIT.2015.7125236) an there are know. Please do not write that this is now solution.

Author response: The novelty of this work is the application of the versatile buck-boost converter as a single-phase rectifier (AC-DC application), which is a new application for the converter proposed while the application realized by F. Mendez-Diaz (2015) is for photovoltaic applications, this last is a DC-DC application. All the works developed to date with the converter have been solely for DC-DC applications, and this is the first work that delves into an AC-DC application. Therefore, we have not emphasized in the introduction that the novelty is the converter topology but an AC-DC application.

Reviewer#2, Concern # 2: At the beginning of the introduction, the authors describe sensorless solutions in the electric drive, but in no part of the text did they indicate that these solutions concern only the lack of a speed sensor or the position of the machine shaft. This is important because the term "sensorless" in electric drives only applies to systems that do not measure angle or speed.

Author response: Thank you very much for the comment. We added a paragraph in the introduction to highlight your observation.

Reviewer#2, Concern # 3: The value of the current depends on the parameters of inductive elements, therefore the authors should note how the quality of the presented solution changes depending on the change in temperature and parameters of inductive elements. Pleas add some comments in text. (http://przyrbwn.icm.edu.pl/APP/PDF/133/app133z4p65.pdf)

Author response: Effectively the inductor value changes with the temperature, but also with the DC current and switching frequency. Therefore, we added sensibility analysis of each converter components for the predictive current control, which depends on the converter parameters in Figure 10, in the new version of the manuscript.

Reviewer#2, Concern # 4: Table 3 schould be push to appendix.

Author response: Thank you very much for noticing.  We eliminated this table according to the  recommendation of reviewer 1 and added a text about the thermal impedance model, selected from the datasheet of the power device.

Reviewer#2, Concern # 5: The sentence "Figure 7 shows that the converter output voltage ...." should be more precise and contain information about which version of the layout it refers to. The note also applies to the description of Figure 7.

We have improved all the figures to enhance their viewing and understanding.

Reviewer#2, Concern # 6: Please explain more clear to the reader why the efficiency of the presented systems changes over time in the presented figure 9. Please add a comment.

Author response: The shown AC-DC conversion results correspond to over a one period of the grid signal (20 ms), for that reason the efficiency changes with the time for each operating point due to the input voltage has variations.  We added information in the Figure and grouped all the related figures in the time domain in Figure 7 (new manuscript version) and added text with more description about these results.

Reviewer 3 Report

DC Voltage Sensorless Predictive Control of a High-Efficiency PFC Single-Phase Rectifier based on the Versatile Buck-Boost Converter

The titled study is interesting both from fundamental and technical point of views. However, the introduction of the paper is too long. It should be reduced to a maximum of one page or so. The picture quality, especially fonts, provides in some figures are no good, thus improvement is necessary. The conclusion reached matches with the discussion made of the paper. The basic discussions made are sound so I am fine with it. 

Author Response

 Reviewer#3, Concern # 1 The titled study is interesting both from fundamental and technical point of views. However, the introduction of the paper is too long. It should be reduced to a maximum of one page or so. The picture quality, especially fonts, provides in some figures are no good, thus improvement is necessary. The conclusion reached matches with the discussion made of the paper. The basic discussions made are sound so I am fine with it.

Author response:  Thank you very much for your comment. It was not possible to reduce the introduction since another reviewer required added information. We have improved all the figures to enhance their viewing and understanding.

Round 2

Reviewer 1 Report

 All the errors and misunderstandings have been corrected, so it is ready to be published. Congratulations for the excellent work!